# Fossil Fruits of *Ceratophyllum* from the Upper Eocene and Miocene of South China

**DOI:** 10.3390/biology11111614

**Published:** 2022-11-04

**Authors:** Shenglan Xu, Hanzhang Song, Helanlin Xiang, Weiqiu Liu, Cheng Quan, Jianhua Jin

**Affiliations:** 1State Key Laboratory of Biocontrol and Guangdong Provincial Key Laboratory of Plant Resources, School of Life Sciences/School of Ecology, Sun Yat-sen University, Guangzhou 510275, China; 2State Key Laboratory of Palaeobiology & Stratigraphy, Nanjing Institute of Geology & Palaeontology, CAS, Nanjing 210008, China; 3School of Earth Science and Resources, Chang’an University, Xi’an 710065, China

**Keywords:** *Ceratophyllum*, Eocene, Miocene, fruit fossil, palaeophytogeography, South China

## Abstract

**Simple Summary:**

Two fruit fossil species of *Ceratophyllum* L. are discovered from South China, namely *C.* cf. *muricatum* Chamisso from the upper Eocene of the Maoming Basin, Guangdong, and *C. demersum* L. from the Miocene of the Guiping Basin, Guangxi. Our findings provide evidence for the distribution of *Ceratophyllum* in South China in the late Eocene, and its wide expansion in subtropical China during the Miocene.

**Abstract:**

*Ceratophyllum* L. is a cosmopolitan genus of perennial aquatic herbs that occur in quiet freshwaters. Fossils of this genus have been widely reported from the Northern Hemisphere, most of them occurring in the temperate zone. Here, we describe two species of fossil fruits discovered from subtropical areas of China. The fossil fruit discovered from the upper Eocene Huangniuling Formation of the Maoming Basin is designated as *C*. cf. *muricatum* Chamisso, and fruits discovered from the Miocene Erzitang Formation of the Guiping Basin are assigned to the extant species *C*. *demersum* L. The discovery of these two fossil species indicates that *Ceratophyllum* had spread to South China by the late Eocene and their distribution expanded in subtropical China during the Miocene.

## 1. Introduction

Ceratophyllaceae Gray is a family of submersed, hydrophilous, perennial, and herbaceous plants. It consists of only one cosmopolitan genus, *Ceratophyllum* L., and about six extant species [1]. Ceratophyllaceae is a sister group of the eudicots [2]. The infrageneric taxonomy of *Ceratophyllum* is largely based on fruit characters. It is generally divided into three sections, with two species in each section. Sect. *Ceratophyllum* has fruits with three to five long spines, sect. *Muricatum* has fruits with spiny and sometimes winged margins with stylar and basal spines and sect. *Submersum* has fruits with a spiny margin without stylar and basal spines [3,4]. 

Ceratophyllaceae is estimated to have diverged from the rest of the eudicots in the Early Cretaceous based on a time-calibrated global angiosperm phylogeny [5]. The earliest fruit fossil convincingly assigned to *Ceratophyllum* was reported from the Upper Cretaceous of Mexico, namely *C. lesii* Estrada-Ruiz, Calvillo-Canadell et Cevallos-Ferrizis [6]. *Ceratophyllum* has only one fossil record from the Paleocene: *C. furcatispinum* Herendeen, Les et Dilcher from Fort Union Formation, Montana, USA [7]. *Ceratophyllum* fossils have been recorded from the Eocene of North America and China, which are assigned to *C. muricatum* subsp. *incertum* (Berry) Herendeen, Les et Dilcher, and *C.* aff. *muricatum* Cham. [7,8]. Oligocene to Pleistocene fossil fruits are relatively abundant and widely distributed in the Northern Hemisphere, including North America, Europe, and Asia [9,10,11,12,13,14,15,16,17]. 

In this paper, two species of *Ceratophyllum* fruit fossils are described from the late Eocene and Miocene strata of South China. The new fossil occurrences provide important insights into the palaeophytogeography of this genus.

## 2. Materials and Methods

The fossil specimens investigated here were collected from two localities within South China (Figure 1). One specimen was collected from the upper part of the Huangniuling Formation (21°42′33.2″ N, 110°53′19.4″ E) of the Maoming Basin, Guangdong. This formation is composed mainly of gray, yellow to white sandstones, siltstones, and conglomerates [18]. The age of Huangniuling Formation is late Eocene according to palaeomagnetic data [19] and pollen assemblages [20]. Other fruit specimens were collected from the Erzitang Formation (23°23′09.67″ N, 110°09′55.21″ E) of the Guiping Basin, Guangxi. The Erzitang Formation is mainly composed of greyish yellow and red mudstone. The geological age of this formation is Miocene based on the mammal fossil *Prolipotes yujiangensis* Zhou, Zhou et Zhao [21,22]. 

Specimens were photographed and measured using a Sony Alpha 6400 Camera and a Nikon SMZ25 stereo microscope. The pericarp of the fossil fruit was examined using a JSM–6330F scanning electron microscope. The terminology for *Ceratophyllum* fruit description follows that used in the monograph of Les [3]. All fossils described here are deposited in the Museum of Biology, Sun Yat-sen University, Guangzhou.

## 3. Results

### 3.1. Ceratophyllum cf. muricatum *Chamisso*

Family: Ceratophyllaceae Gray

Genus: *Ceratophyllum* L.

Section: *Muricatum* Les

Species: *Ceratophyllum* cf. *muricatum* Chamisso

Specimen: MMJ3–2907

Locality: Maoming Basin, Guangdong, South China

Geological horizon and age: Huangniuling Formation, late Eocene

Description: Achene axis symmetric, elliptic to suborbicular, 3.6 mm long, 3.2 mm wide (excluding the spines), with a length/width ratio of about 1.1 (Figure 2A,C). The surface of the fruit has granular ornamentation (Figure 2B). The fruit body has five prominent spines: one stylar spine (incomplete, ca. 0.9 mm long), two lateral spines 0.9–1 mm long, and two basal spines 2.7–3.8 mm in length (Figure 2A,C). The impression of a marginal wing is clear, and slightly toothed at the apex (Figure 2A).

Remarks: The characteristics of *Ceratophyllum* fruit include shape, size, spine, surface, and wings [3], while the most informative taxonomic characters are the fruit size and the spines [25]. The morphological comparison of our fossil to other *Ceratophyllum* fruit fossils is summarized (Table 1).

The fossil fruit from Maoming is characterized by it being an axially symmetrical fruit with marginal wings and five spines: a stylar spine, two lateral spines and two basal spines. Accordingly, this fruit can be assigned to sect. *Muricatum*. This section includes two modern species, *C. tanaiticum* Sapjegin and *C. muricatum* Chamisso.

The fossil from Maoming in compared to *Ceratophyllum tanaiticum* has fewer lateral spines. *C. tanaiticum* is a Pontic-Caspian endemic relict plant [3]. Fruit of *C. tanaiticum* possess several short lateral and basal spines, with the length of spines about 0.35–3.15 mm and a minute stylar spine [4,26,27]. Meanwhile, our fossil has only two clear lateral spines and longer basal spines.

Instead, our fossil is more morphologically similar to *Ceratophyllum muricatum. C. muricatum* have at least two basal spines of achene margin, and fruit body length is less than 4.5 mm [27]*. C. muricatum* has three subspecies, among which, fruits of *C. muricatum* subsp. *australe* (Grisebach) Les and subsp. *muricatum* Cham. have more lateral spines compared with those of subsp. *kossinskyi* (Kuzen.) Les [4]. Our fossil fruit can be easily distinguished from previously reported fossil species assigned to sect. *Muricatum* based on the number of lateral spines: our fossil fruit has only two lateral spines, while those reported fossil species have at least five lateral spines [7,8,14]. Despite minor intraspecific variation, there is very small amount of evolutionary change in *Ceratophyllum* [28]. Therefore, we prefer to assign the fossil to *Ceratophyllum* cf. *muricatum* Chamisso.

### 3.2. Ceratophyllum demersum *L.*

Family: Ceratophyllaceae Gray

Genus: *Ceratophyllum* L.

Section: *Ceratophyllum* L.

Species: *Ceratophyllum demersum* L.

Specimens: GP427–GP456

Locality: Guiping Basin, Guangxi, South China

Geological horizon and age: Erzitang Formation, Miocene

Description: The fruit bodies (excluding spines) are ovate, slightly broad at the apex, 3.42–4.68 mm long and 2.09–2.80 mm wide (Figure 3A,B), with a length/width ratio of 1.48–1.79. The surfaces of the fruit bodies are tuberculate, with striated grooves near the edge (Figure 3A,E,G,I,J). The achenes have an apical stylar spine and a pair of basal spines (Figure 3A). The largest stylar spine is 2.79 mm in length and 0.33 mm in width (Figure 3K). The length of stylar spines ranges from 0.22 to 2.79 mm, with a mean of 0.76 mm. The length of basal spines ranges from 0.15–1.35 mm, with a mean of 0.59 mm. Cotyledons are observed when the fruits are longitudinally split into two halves (Figure 3D,F,H). Two symmetrical holes are present on the base, which represent the pedicel connection (Figure 3C). There is a small projecting part at the base (Figure 3G,L). 

The outer surface of the fruit exocarp consists of irregular polygonal cells with lengths of 10–25 μm (Figure 4A,B,E), with some of the cells having parallel partitions (Figure 4C,D). The inner surface of the endocarp is composed mainly of rectangular cells, about 20 μm × 40 μm in size (Figure 4F). 

Remarks: Our fossils have three spines, two basal spines, and one stylar spine. The fruits have a rough surface and no marginal wings. These characteristics are consistent with the fruit features of sect. *Ceratophyllum* (fruits with three to five spines but no winged margins). Sect. *Ceratophyllum* includes two species: *C. platyacanthum* Chamisso and *C. demersum* L. The fruit of *C. platyacanthum* has five spines, which include two distinctive facial spines. Our three-spined fruit fossils can be distinguished from this species. In comparison, the fruit of *C. demersum* is 3.5–6 × 2–4 mm in size, with two basal spines and one stylar spine, lacks a marginal wing, and its facial spines are absent. 

By carefully comparing the shape, size, and spines of our specimens and the fruit characters of the extant species *C. demersum*, we cannot discern any significant morphological differences between them. Additionally, the morphological data of our fossil materials is close to other fossils designated as *C. demersum* [11,17] (Table 1). So, these fossil fruits are assigned to *C. demersum* L. 

## 4. Discussion

Species of the genus *Ceratophyllum* are known to be highly tolerant of a wide range of temperature and salinity values, leading to their broad distribution (Figure 5). The aquatic environment in which representatives of this genus live provides relatively buffered conditions minimizing pressure on its evolution and distribution [30]. 

Fossils of *Ceratophyllum* have been widely reported from the Northern Hemisphere (Figure 5). The earliest fossil *C. lesii* was found from the Upper Cretaceous of Mexico [6] and is similar to the modern species *C. demersum*. An extinct genus *Donlesia* Dilcher et Wang with affinities to Ceratophyllaceae was reported from the Lower Cretaceous of Kansas, USA [31,32]. Based on these findings, it is considered that Ceratophyllaceae may have originated in North and Central America during the Cretaceous, in accordance with the divergence time of Ceratophyllaceae from the rest of the eudicots based on a time-calibrated global angiosperm phylogeny [5].

The earliest *Ceratophyllum* fossil of sect. *Muricatum* was discovered from the Paleocene in the Fort Union Formation in the United States, designated as *C. furcatispinum* [7]. Later, *C. muricatum* subsp. *incertum* was discovered from the lower and middle Eocene in North America [7], and *C.* aff. *muricatum* from the middle Eocene of China [8]. The two species share a great number of morphological similarities. The discovery of fossils of *Ceratophyllum* in China suggests a floristic exchange between Asia and North America during the Eocene [8]. Our discovery of *C.* cf. *muricatum* from the upper Eocene of Maoming Basin provides further evidence supporting this supposition, which also suggests that *Ceratophyllum* was distributed in the subtropical region at that time. 

*Ceratophyllum* fossils have also been reported from Asia in the Oligocene, such as *C. zaisanicum* Avakov from the Zaysan Basin of Kazakhstan [33], *C. submersum* L., and *C. tenuicarpum* Dorof. from Siberia, Russia [10,11]. The distribution area of this genus expanded during the Miocene, and *Ceratophyllum* fossils have been widely reported in Asia and Europe [11,12,16,17], such as the fruit fossils *C. miocenicum* Dorof. and *C. pannonicum* Dorof. from the Orlovka and Lgov, Russia, respectively [11], and *C. lusaticum* Mai from Leipzig, Germany [12]. From the middle Miocene Shanwang Formation of eastern China, stems and fruits referred to *Ceratophyllum* have been reported [9,14]. The discovery of Miocene fruit fossils of *C. demersum* from the Erzitang Formation of the Guiping Basin, Guangxi, together with the discovery of the same species from the upper Miocene of Huaning County, Yunnan, Southwest China [17], suggest that *C. demersum* was widely distributed in subtropical China in the Miocene.

## 5. Conclusions

Two fruit fossil species of *Ceratophyllum* are reported from low latitude of Asia in this paper. By morphological comparison with modern and fossil species, fruit fossil found from the upper Eocene Huangniuling Formation of the Maoming Basin, Guangdong, is assigned to *C*. cf. *muricatum* Chamisso. The discovery of this fossil indicates that *Ceratophyllum* has been distributed in South China by the late Eocene. Fossils discovered from the Miocene Erzitang Formation of the Guiping Basin, Guangxi, are designated to the extant species *C. demersum* L. The emergence of these fossils, together with the discovery of the same species from the upper Miocene of Yunnan, confirms the wide distribution of this genus in the subtropical China during the Miocene.

## Figures and Tables

**Figure 1 biology-11-01614-f001:**
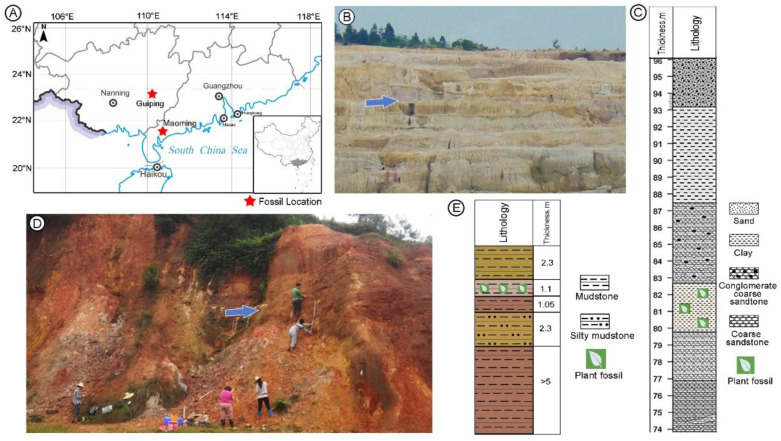
The geographic location, and stratigraphic sections of two fossil sites, one in the Maoming Basin and one in the Guiping Basin, South China. (**A**) Geographical location map of the Maoming and Guiping basins. (**B**) Field photo of lithological characteristics of the Huangniuling Formation, Maoming, the blue arrow indicates the fossil layer. (**C**) Stratigraphic column of the outcrop section in B, the upper part of Huangniuling Formation (modified from Herman et al. [23]). (**D**) Field photo of lithological characteristics of the Erzitang Formation from Guiping, the blue arrow indicates the fossil layer. (**E**) Stratigraphic column of the outcrop section in D (modified from Huang et al. [24]).

**Figure 2 biology-11-01614-f002:**
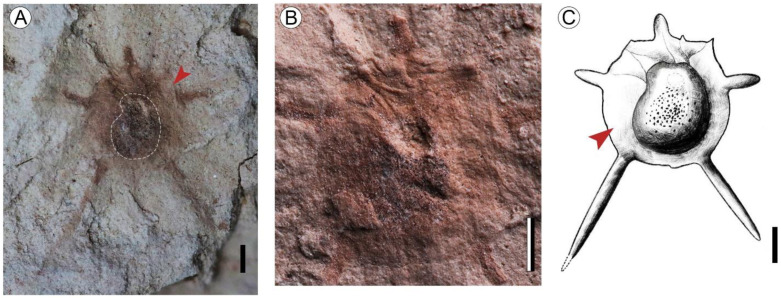
Fossil and reconstruction image of *Ceratophyllum* cf. *muricatum* Chamisso from the Maoming Basin. (**A**) Fossil fruit of *C.* cf. *muricatum* with five prominent spines, red arrow shows marginal wing, the dotted line shows the position of seed, MMJ3–2907. (**B**) Enlargement of A, the middle part of the fruit exhibits a granular surface texture. (**C**) Fossil reconstruction, red arrow shows marginal wing. Scale Bars = 1 mm.

**Figure 3 biology-11-01614-f003:**
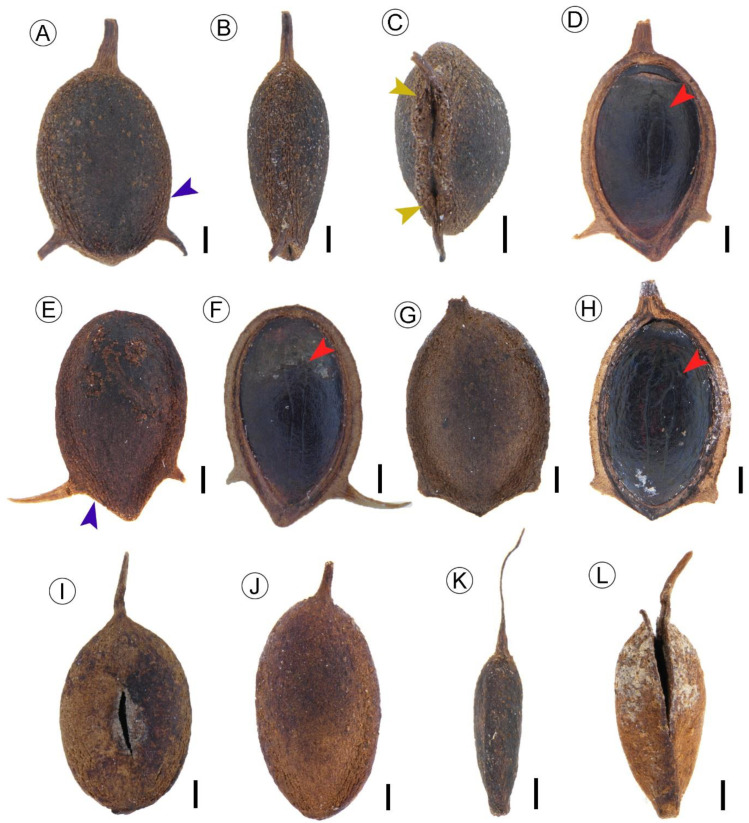
Fruit fossils of *Ceratophyllum demersum* L. from the Guiping Basin. (**A**–**C**) GP427. The blue arrow shows striated grooves near the edge, yellow arrows show holes at the base. (**D**) GP428a. Inner view of fruit, the red arrow shows the trace of a cotyledon. (**E**,**F**) GP428b. The blue arrow indicates the grooves on the surface, the red arrow shows the trace of a cotyledon. (**G**) GP429a Fruit outer surface, showing the location of spines. (**H**) GP429b. Inner view of fruit, red arrows indicate cotyledon traces. (**I**) GP430. Fruit with a stylar spine. (**J**) GP432. Fruit with an irregular tuberculate surface. (**K**) GP436. Lateral view of the fruit, showing a long stylar spine. (**L**) GP439. Lateral view of the dehiscent fruit. Scale Bars = 0.5 mm.

**Figure 4 biology-11-01614-f004:**
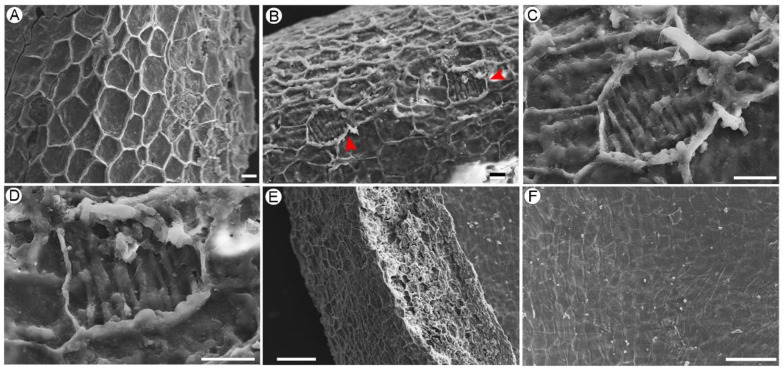
Scanning electron microscopic image of the pericarp of *Ceratophyllum demersum* L., GP440. (**A**–**D**) The outer surface of the exocarp with polygonal cells, red arrows show the parallel partitions. (**E**) Lateral view, showing the fiber structure of the pericarp. (**F**) The inner surface of the endocarp with rectangular cells. Scale Bars = 20 μm (**A**,**B**), 10 μm (**C**,**D**), 100 μm (**E**,**F**).

**Figure 5 biology-11-01614-f005:**
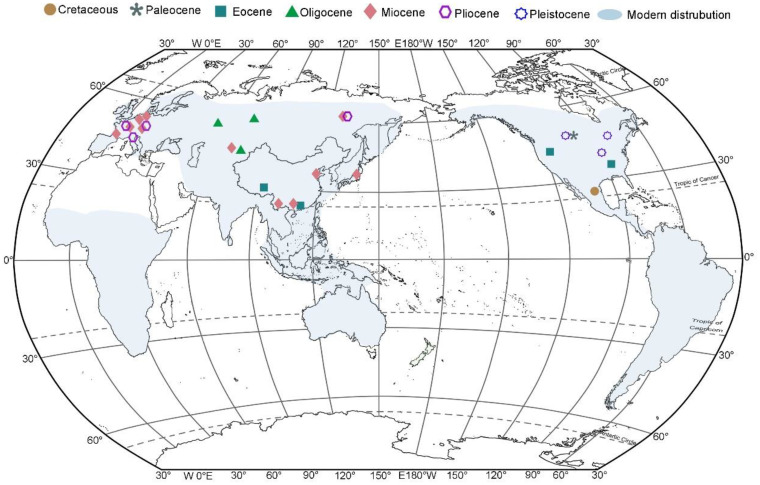
Geological distribution map of *Ceratophyllum* modern species, and fossil records.

**Table 1 biology-11-01614-t001:** Morphological comparison of two fruit fossils *Ceratophyllum* cf. *muricatum* Chamisso and *C. demersum* L. with other *Ceratophyllum* fruit fossils.

Species	Achene Shape	Length (mm)	Width (mm)	L/W Ratio	Surfaces Ornament	Length of Stylar Spine (mm)	Length of Basal Spine (mm)	Facial Spines	Marginal Wing	Lateral Spines	Occurrence	Age	References
*Ceratophyllum* cf. *muricatum*	elliptic to suborbicular	3.6	3.2	1.1	granular	0.9	2.7–3.8	−	+	2	Guangdong, China	late Eocene	This paper
*C. demersum*	ovate	3.42–4.68	2.09–2.80	1.48–1.79	tuberculate, with striated grooves	0.22–2.79	0.15–1.35	−	−	−	Guangxi, China	Miocene	This paper
*C. lesii*	elliptical	2.1	1	2.1	smooth	incomplete	4.2	−	+	−	Northern Mexico	Late Cretaceous	[6]
*C. furcatispinum*	/	4.8	1.8	2.7	/	2.3	/	−	+	9–10	Montana, USA	Paleocene	[7]
*C. muricatum* subspecies *incertum*	elliptical	3.4	2.2	1.65	smooth	1.8	3.3	−	−	8–11	Tennessee, USA	Eocene	[7]
*C.* aff. *muricatum*	ellipsoidal	3	2.2	1.4	smooth	1.5	/	−	−	12	Tibet, China	middle Eocene	[8]
*C. zaisanicum*	elliptic	4.0	2.7	1.48	finely rugose	/	/	−	−	9–19	Kazakhstan	early Oligocene	[11]
*C. tenuicarpum*	oblong	/	/	/	/	/	/	/	narrow	+	Siberia, Russia	Oligocene	[11]
*C. submersum*	elliptic	2.7–3.5	1.7–2.6	1.44	smooth, rarely slightly rugose	/	/	−	−	−	Rostov, Russia; Odessa, Ukraine	Oligocene –Pliocene	[11]
*C. echinatum*	elliptical	5.0	3.6	1.6	not sure	4.1	3.1	−	+	10–11	Nevada, USA	Miocene	[7]
*C. spinulosum*	wide-elliptic	/	/	/	/	−	−	−	wide	+	Siberia, Russia	Miocene	[11]
*C. miocenicum*	obovate	2.0–2.1	1.8–1.9	1.11	smooth or slightly rugose	−	−	−	narrow, toothed	−	Rostov, Russia; Odessa, Ukraine	Miocene	[11]
*C. tanaiticum*	elliptic	/	/	/	/	/	/	/	fimbriate	−	Ukraine	Miocene	[11]
*C. pannonicum*	elliptic	3.0–3.8	2.0–2.7	1.45	finely rugose	+	/	/	wide	10–18	Ukraine	Miocene	[11]
*C. pannonicum*	/	3 (without stylus)	2.3	1.3	tuberculate	/	/	+	−	4–5	Klettwitz, Germany	Middle Miocene	[12]
*C. sinjanum*	/	6–8	2.5–3	/	/	/	/	/	/	/	Sinj, Croatia	Middle Miocene	[29]
*C. demersum*	ovate to elliptical	3.9 ± 0.3	2.9 ± 0.5	1.3	papillae or are smooth	5.9	5.2	−	−	−	Yunnan, China	Late Miocene	[17]
*C. demersum*	elliptic	3.6–4.1	2.1–2.8	1.6	rugose	+	+	−	−	−	Siberia, Voronezh, Rostov, Russia	Miocene, Pliocene	[11]
*C. tenuicarpum*	oblong oval	3.1–4.9	1.6–2.5	/	/	/	/	−	/	−	Siberia, Russia	Miocene to Pliocene	[10]
*C. protanaiticum*	oblong	/	/	/	/	−	−	−	wide, with irregular margin	+	Voronezh Region, Russia	Pliocene	[11]
*C. platyacanthum*	oblong	/	/	/	/	/	/	/	+	+	Bashkortostan, Russia	Pliocene	[11]

/: No description; (+): present; (−): absent.

## Data Availability

All data dealing with this study are reported in the paper.

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
