# Peer review of "Fossil Fruits of Ceratophyllum from the Upper Eocene and Miocene of South China"

_biology, 2022, doi:10.3390/biology11111614_

Round 1
Reviewer 1 Report
It is a clear, well-structured manuscript. It brings new information on two taxa C. cf. muricatum subsp. kossinskyi (Kuzen.) Les from the upper Eocene Huangniuling Formation of the Maoming Basin and C. demersum L. from the Miocene Erzitang Formation of the Guiping Basin. I recommend the manuscript for publication after minor revision.
A list of recommended corrections is attached here:
Page 3 line 93: You say, the fossil fruit … is characterized by … marginal wings … Please add an arrow to figure 2 showing the character.
Page 3 line 96: I would recommend starting the sentence with "Compared to..." or "Fossil from Maomin in compare to ... "
Page 4 line 133: You say, holes are present on the base. Please add an arrow to figure 3C showing the character.
Legend to figure 3 line 155: „(E–F) GP428b, the blue arrows“ there is only one blue arrow on picture 3E, so please change the arrows to arrow
Conclusion line 209: C. cf. muricatum subsp. kossinskyi (Kuzen.) Les. Usually italica is used for taxon names.
Conclusion line 212: C. demersum L. Usually italica is used for taxon names.
Reviewer 2 Report
The present manuscript is a short but interesting paper, describing two species of Ceratophyllum (Ceratophyllaceae): C. cf. muricatum subsp. kossinskyi (Kuzen.) Les from the late Eocene of the Maoming Basin, and C. demersum L. from the Miocene of Guiping Basin, South China. These new findings have important evolutionary and phytogeographic implications. Ceratophyllaceae morphologically demonstrates no close affinities with any other extant group of angiosperms, which suggests that the family may be a vestige of an ancient line of angiosperms (Les, 1988; Taylor, 2009). Meanwhile fossils within this family always demonstrating high morphological stasis, as is also seen in the present fossils. Furthermore, this study also expands the paleogeographic distribution of Ceratophyllum. The text of the manuscript is well organized. The results and discussion are accurately presented. The images are in high-quality. I recommend it for publication with minor revision.
Specific comments and suggestions are as follows:
1. The assignment of a late Eocene fruit to a modern subspecies is controversial, because the variation of the number of lateral spines and their length is not clear based on a single specimen, while these characters are used to distinguish the subspecies of Ceratophyllum muricatum. Besides, the occurrence of extant species of Ceratophyllum in the Paleogene (Herendeen et al., 1990) is unusual, which is interpreted as exhibiting very small amount of morphological change in this genus, though (Taylor et al., 2009). Thus, it is more proper to refer this fruit to Ceratophyllum cf. muricatum.
Taylor, T.N.; Taylor, E.L.; Krings, M. Paleobotany: The Biology and Evolution of Fossil Plants (Second Edition); Elsevier/Academic Press: Burlington, MA, London, San Diego, CA, and New York, NY, 2009.
Herendeen, P.S.; Les, D.H.; Dilcher, D.L. Fossil Ceratophyllum (Ceratophyllaceae) from the Tertiary of North America. Am. J. Bot. 1990, 77(1), 7–16.
2. For systematic part, put the sections Muricatum Les and Ceratophyllum L. between lines 72 and 73, 115 and 116, respectively.
3. Table 1: put occurrence and age columns just before the reference column.
4. Figure 3: legends need to be improved, such as changing “GP428a, inner view of fruit, red arrow shows the trace of a cotyledon.” to “GP428a. Inner view of fruit showing the trace of a cotyledon indicated by a red arrow.” You should always put a dot after specimen number.
5. Small linguistic issues are as follows:
Line 11–20, 72, 73, 115, 116: genus and species names should be in italic format;
Line 17: keep one dot after “C. demersum L”;
Line 36: delete “the Paleocene”;
Line 36, 38: change “&” to “et”;
Line 58, 72, 83, 91, 95, 115, 169, 199: correct “Cerataphyllum” as “Ceratophyllum”;
Line 82: add “, and slightly toothed at the apex” after “marginal wing is clear”;
Line 83, 134: change “Systematic affinity within Cerataphyllum” to “Remarks”;
Line 98: correct “species” as “subspecies”;
Line 121: change “by” to “and”;
Line 142, 143: cancel bold format;
Line 171: correct “Donlesia Dilcher and Wang” as “Donlesia Dilcher et Wang”;
Line 186: change “only” to “also”;
Line 189: add “and” before “Ceratophyllum fossils”;
Line 201: insert “including” before “C. cf. muricatum”.
